# Actin Mutations and Their Role in Disease

**DOI:** 10.3390/ijms21093371

**Published:** 2020-05-10

**Authors:** Francine Parker, Thomas G. Baboolal, Michelle Peckham

**Affiliations:** School of Molecular and Cellular Biology, University of Leeds, Leeds LS2 9JT, UK; f.parker@leeds.ac.uk (F.P.); Thomas.baboolal@gmail.com (T.G.B.)

**Keywords:** actin, mutation, polymerization, myosin

## Abstract

Actin is a widely expressed protein found in almost all eukaryotic cells. In humans, there are six different genes, which encode specific actin isoforms. Disease-causing mutations have been described for each of these, most of which are missense. Analysis of the position of the resulting mutated residues in the protein reveals mutational hotspots. Many of these occur in regions important for actin polymerization. We briefly discuss the challenges in characterizing the effects of these actin mutations, with a focus on cardiac actin mutations.

## 1. Introduction

Actin is a globular protein (G-actin) that assembles into filaments (F-actin) and is important for cell movement, intracellular movement, muscle contraction and many other functions. There are six actin genes in the human genome. Three of these encode the α-actin isoforms found in cardiac, skeletal or smooth muscle (*ACTC1, ACTA1* and *ACTA2*, respectively). Two encode γ-actin, of which one is widely expressed (*ACTG1*) and the other is smooth muscle specific (*ACTG2*). The final gene encodes the widely expressed β-actin (*ACTB*). These actin isoforms are highly (>90%) conserved at the protein level. Actin is a promiscuous protein, interacting with many other proteins [1], and is also subject to many different post-translational modifications [2].

The structure of actin has been solved multiple times in its monomeric [3] and, more recently, in its filamentous form, the latter building on advances in cryo-electron microscopy [4,5,6,7,8]. In addition, there are structures of F-actin in the complex with a myosin motor domain, and/or with other F-actin filament binding proteins such as cofilin and tropomodulin [9,10]. These structures show that the actin monomer is divided into two halves (inner and outer domains) by a cleft that binds nucleotide and cation (Mg^2+^). Each half is further divided into two domains, with one domain comprising subdomains 1 and 2, and the other subdomains 3 and 4. Subdomains 1 and 2 are found on the outer edge of the actin filament. In the actin filament, each monomer makes interactions longitudinally within a protofilament (along the actin filament), as well as laterally (between the two protofilaments), such that each actin monomer (subunit) interacts with its three surrounding subunits. The currently available structures of G- and F-actin provide a rich resource for understanding how myosin interacts with actin, how the actin monomer forms filaments and interacts with a variety of actin binding proteins and how disease-causing mutations in actin affect its biological function.

Disease-causing mutations have been reported for each of the six actin genes, demonstrating the importance of actin for normal cell behaviour and function in a variety of cell types. The majority of these (>90% for five out of the six actin genes) result in missense mutations in the protein (Appendix A), and typically, these mutations are dominant. The type of disease that mutations in a specific actin gene commonly cause reflect its expression pattern, as detailed below. Moreover, mutations occur throughout the entire sequence for each actin gene. It is therefore of interest to determine if there are any specific residues or common functional regions in the encoded proteins in which these missense mutations are found. The main aim of this article is to evaluate these missense mutations, determine if there are any mutational ‘hotspots’ and the potential consequences for actin function. We then go on to discuss how a range of approaches is needed to test experimentally what these consequences are, and how they might lead to disease, with a specific focus on cardiac actin mutations.

## 2. Disease-causing Mutations in the Six Actin Genes, an Overview

The gene with the highest number of reported mutations (over 220) is *ACTA1*, which encodes the isoform of α-actin almost exclusively expressed in skeletal muscle. Mutations in *ACTA1* are found throughout the sequence and >92% result in single amino acid substitutions in the protein (Figure 1 and Appendix A). They are a common cause of Nemaline Myopathy (reviewed in [11,12]), a non-progressive skeletal muscle disease that commonly has an early onset, with severe cases diagnosed at birth. This disease typically results in muscle weakness, particularly in the respiratory muscles, which can cause breathing difficulties, but also difficulties in swallowing in severe cases.

The second most commonly mutated gene is in *ACTA2* with over 80 mutations, of which 93% are missense (Figure 1 and Appendix A). *ACTA2* encodes an α-actin isoform that is highly expressed in specific smooth muscle cells associated with the vasculature. Perhaps not unsurprisingly, mutations in this gene are strongly associated with familial thoracic aortic aneurysms, and *ACTA2* is the most frequent gene mutated in this disorder [13]. In rarer cases, mutations in *ACTA2* cause cerebral arteriopathy, in which the most commonly affected residue is Arg179 [14].

Mutations in *ACTC1*, the gene that encodes the isoform of α-actin found predominantly in cardiac muscle, are the next most common, with over 70 mutations, of which over 90% are missense (Figure 1 and Appendix A). Mutations in this gene cause heart disease [15]. Over 50% of the known mutations in *ACTC1* cause hypertrophic cardiomyopathy (HCM) and about 20% cause dilated cardiomyopathy (DCM). The remainder cause left ventricular non-compaction or other heart defects. *ACTC* is one of eight disease genes that contain missense mutations causing HCM, and one of 20 disease genes causing DCM [16,17]. Mutations in *ACTC* are common in patients with apical HCM (50%) and patients are heterozygous for the mutant allele. HCM affects 1:500 to 1:250 people and is a common cause of premature death in young adults [18,19]. In HCM, typically the left ventricular wall and/or septum between left and right ventricles thickens, and the myocytes (muscle cells) become disorganised. DCM is less common, affecting around 1:3000 people [20]. In DCM, the left ventricular wall becomes thinner. Ventricular relaxation is impaired, restricting the atrial emptying into the ventricles, and resulting in dilation of the atria [21]. The high incidence of sudden death seen in DCM patients results from the impaired systolic function of the left ventricle, which can lead to related pathologies such as thromboembolic events and arrhythmias.

About 70 mutations have been described for *ACTB,* which encodes the widely expressed β-actin (Figure 1, Appendix A). However, in this case only ~50% are missense mutations. Consistent with its widespread expression, mutations result in a broad range of defects, including a specific facial appearance, intellectual disability, hearing loss, heart and renal defects, brain abnormalities, neuronal migration defects and muscle wasting, typical of a syndrome named Baraitser-Winter syndrome [22,23,24]. The remaining mutations in *ACTB* include deletions, premature stop codons and frameshift mutations, of which most (40%) are gross deletions, many of which lead to complete gene loss [25]. The resulting haploinsufficiency of β-actin is associated with widespread effects, including developmental delay, organ malformations, and growth retardation, although this phenotype is considered distinct from the symptoms associated with Baraitser-Winter syndrome [25]. Mutations in *ACTB* have also more recently been associated with bleeding disorders [26].

A second widely expressed protein is γ-actin, encoded by the gene *ACTG1*. Over 50 mutations have been reported for *ACTG1,* of which all but one are missense mutations (Figure 1 and Appendix A). Just over half of these mutations cause deafness [27]. This is consistent with the expression and role of γ-actin in the sensory epithelial cells of the inner ear, in which γ-actin is an essential component of the stereocilia, along with β-actin. Distortion of the stereocilia is essential for the detection of sound. Although stereocilia form normally in a mouse γ-actin knockout model, maintenance of these structures is affected and the mice show a progressive loss of hearing [28]. This research also demonstrated that γ-actin (*ACTG1*) is not strictly required for development, possibly because levels of β-actin increased in the knockout mouse model, partly compensating for the loss of γ-actin. However, a high proportion of the remaining mutations in *ACTG1* cause Baraitser-Winter syndrome, consistent with the widespread expression of γ-actin in multiple tissues [29]. It is not clear why some mutations in *ACTG1* appear to have a more limited effect than others, as affected residues for both types of disease are distributed throughout the sequence (Appendix A).

Mutations in *ACTG2* are the least common, with just over 20 described. Almost all of these (96%) are missense (Figure 1 and Appendix A). The expression of this γ-actin isoform is restricted to smooth muscle cells in the gut, prostate, bladder and adrenal gland. Mutations in *ACTG2* cause Megacystis microcolon-Intestinal hypoperistalsis syndrome [30], also known as chronic intestinal pseudo-obstruction, visceral myopathy (or degenerative leiomyopathy) [31]. All of these are disorders of enteric smooth muscle function. They mainly affect the intestine leading to chronic intestinal obstruction, and can also affect the bladder. The smooth muscle cells in the smooth muscle layers that surround the gut epithelium are important for moving the contents of the gut along its length. The organisation of these smooth muscle cells is often disordered in this disease, likely leading to decreased smooth muscle contraction and the resultant intestinal obstruction [31,32].

## 3. Positional Analysis of Disease-Causing Missense Actin Mutations

An analysis of the missense mutations for each of the actin isoforms shows that, while they are found throughout the sequence, there are common mutational hotspots where the numbers of mutations tend to be higher than elsewhere (Figure 1 and Figure 2a,b). The total number of missense mutations (sum total for mutations, Figure 2a) are somewhat dominated by the large number of mutations in skeletal α-actin. However, removing these mutations from the plot (Figure 2b) reveals similar mutational hotspots. This suggests there are key functional regions in all of the actin isoforms where disease-causing mutations are more likely to result in a phenotype.

A major hotspot for mutations is the DNAse-1 loop (or D-loop) in subdomain 2 (SD2). Its name is derived from its ability to bind to DNAse-1, which inhibits F-actin formation, and this interaction was instrumental in generating the first crystal structure for G-actin [33]. The D-loop is crucial for actin polymerisation and is also the target of many actin binding proteins [3]. These include tropomodulin, which caps the pointed end of actin filaments preventing polymerisation and depolymerisation [10], and cofilin, which severs actin filaments [34]. The D-loop is important for polymerisation as it is involved in both lateral (between the two protofilaments) and longitudinal contacts (along the protofilament) between actin monomers, and its structure is sensitive to the occupancy of the nucleotide binding site. The residues Ser14 and the methylated His73 detect the nucleotide state and transmit that information to the D-loop, which can then move its position [8].

In longitudinal contacts, the D-loop inserts into a cleft in the adjacent actin monomer (magenta, Figure 2). Residues in the D-loop (Gln41, Val43 and Val45) interact with residues in SD1 (Tyr143, Arg372 and Phe375) and SD3 (Leu110, Tyr169, Ala170 and Pro172) of the adjacent actin monomer [5,7]. Residues Arg39 and His40 in the proximal region of this loop are additionally involved in only one of only two lateral (inter-protofilament) contacts with adjacent actin monomers. This occurs through an interaction of these D-loop residues in one monomer with residues in the so-called hydrophobic plug of the second monomer (Arg39 interacts with Glu270, His40 interacts with Ser265 and Gly268) [4,6,7]. The distal part of the D-loop is found towards the outer edge of the filament and is involved in interactions with myosin. For example, Glu570 in non-muscle myosin 2C (NM2C) probably forms an electrostatic interaction with Lys49 in the D-loop [35]. A similar interaction with this residue has also been demonstrated for loop 3 of Myo6 [6].

A closer look at mutations in the D-loop reveals three specific residues (His40, Met47 and Gly48) that are mutated in four out of the six actin isoforms (indicated by the red stars in Figure 1). Gly48 is not apparently directly involved in actin or myosin binding, but substitution of this small amino acid residue is likely to have an overall effect on the flexibility of the D-loop, and thus, indirectly affect actin filament polymerization. His40 is critical for longitudinal contacts between monomers in the filament, as discussed above. Moreover, it has long been known that selective carboethoxylation of this residue inhibits actin polymerization [36]. Mutations in His40 are therefore likely to decrease the numbers of filaments. Its mutation to Tyr in skeletal α-actin, cardiac α-actin and γ-actin cause nemaline myopathy, hypertrophic cardiomyopathy and Baraitser-Winter syndrome, respectively [37,38,39]. Its mutation to Asn in smooth α-actin causes thoracic aortic aneurysms and dissections [40] (Appendix A).

Met47 is one of two residues (Met44 and Met47) in the D-loop that are oxidised by MICAL (molecule interacting with CasL). The resulting oxidation causes rapid and catastrophic depolymerisation of the actin filament [41,42] and the resulting monomers do not polymerise as efficiently as non-modified actin monomers. Thus, oxidation of Met44 and Met47 is an alternative strategy for regulating actin polymerisation, in addition to the action of actin severing proteins such as cofilin [43]. Mutation of Met47 to Leu abolishes a longitudinal actin-actin M37-O-T351 contact, and prevents this catastrophic filament disassembly [42]. Met47 is mutated in all three α-actin isoforms and in β-actin. Disease mutations (Appendix A) in this residue that prevent its oxidation might therefore be expected to stabilise actin filaments. In non-muscle cells, filament remodelling and turnover is important for cell motility, and thus, stabilization of β-actin filaments would be expected to affect the behaviour and migration of these cells. However, while mutation of Met47 to Thr in β-actin is known to cause Baraitser-Winter syndrome [24], this disease mutation has not yet been tested to determine if or how it affects actin filament dynamics in cells.

The effects of disease mutations in Met47 in muscle specific α-actin isoforms is less clear. Typically, actin in striated muscle tends to be stably incorporated into filaments. The rate of actin synthesis and turnover in cardiac and skeletal muscle is relatively slow (occurring over weeks [44]). Moreover, the roles of MICAL in striated muscle are also not well understood. In cardiac muscle, mutation of Met47 to Leu in cardiac α-actin causes hypertrophic cardiomyopathy [45]. However, while MICAL3 is expressed in the heart [46], if or how this plays a role in actin polymerization in the heart or in filament maintenance is unclear. Mutation of Met47 to Val in skeletal α-actin causes nemaline myopathy ([47], Appendix A). Although mutations in MICAL cause contractile muscle filaments to become disorganised in skeletal muscle in *Drosophila* [48], this pathway and its role in actin polymerization has hardly been explored in mammalian skeletal muscle. In smooth α-actin, mutation of Met47 causes thoracic aortic aneurysms [49]. Smooth muscle also expresses MICAL, but its role in actin organization in smooth muscle has not been investigated. In addition to the potential interaction with MICAL, and downstream effects on filament polymerization, it is also possible that a mutation in Met47 simply alters the structure of the D-loop and thus destabilises the actin filament. Clearly, effects of mutations in this residue need exploring further.

Two further mutational hotspots are found in the two loops that make a longitudinal interaction between two actin monomers along a protofilament (Figure 2a–d). One loop comprises residues 240-249 in SD4 and the other, residues 321–324 in SD3. The residues Pro243, Asp244 and Gly245 in the loop in SD4 (coloured blue in Figure 2a,c,d) interact with residues Pro322 and Met325 in a second loop in SD3 (coloured yellow in Figure 2a,c,d), and with residues Met283, Ile287 and Asp288 in an adjoining loop in SD3 (not highlighted in Figure 2) [7]. Almost all of these residues are mutated in one or more isoforms (Figure 1 and Appendix A). A mutation in the adjacent acidic residue Glu241, to a basic residue (Lys) in the SD4 loop, is found in smooth and skeletal α-actin, and in γ-actin, causing thoracic aortic aneurysms, nemaline myopathy and deafness, respectively [50,51,52] (Appendix A). This residue interacts with Ser323 and Thr324 in the adjacent actin subunit [7], and mutations are likely to destabilise the actin filament.

A fourth hotspot is in the Thr-rich loop and its flanking helices H5 and H6. The numbers of mutations in this region, across all actin isoforms, are particularly high. Interestingly, not only is this region important in actin-actin interactions in the filament, but specific residues co-ordinate phosphate and Mg^2+^ ions (Figure 1), and this co-ordination is also important for filament stability [7]. The structure of H5 and the Thr-rich loop changes between G- and F-actin [7] and the Thr-rich loop becomes involved in longitudinal contacts between actin monomers. Arg210 in H6 is mutated in four out of the six actin isoforms; to Cys in γ-actin (causing deafness) [53], to Asn in smooth γ-actin and smooth α-actin (causing visceral myopathy and thoracic artery disease, respectively) [54,55] and to His in cardiac α-actin (causing HCM and DCM) [56] (Appendix A). While this residue has not been identified as part of a specific interaction, it must play a critical role in actin-actin interaction in some way. From investigating the orientation of the side-chain rotamers for Arg210 and a glutamate residue downstream (Glu214) in the actin structure, it seems likely that these two residues could form an ionic interaction (i, i+4), which would help to stabilise this helix, as seen in single α-helical (SAH) domains [57].

Two further highly mutated residues include Arg256 found in a helix in subdomain 4 and Pro70 found at the start of the sensor (His73) loop at the boundary between SD2 and SD1. Arg256 is mutated in four out of six actin isoforms: to His or Cys in γ -smooth actin [30,58], to His in smooth α-actin [59], to His or Gly in skeletal α-actin [60] and to Trp in γ-actin [22]). These cause Megacystis microcolon-intestinal hypoperistalsis syndrome, or visceral myopathy (smooth γ-actin), aortic disease (smooth α-actin), nemaline myopathy (skeletal α-actin) and Baraistser-Winter Syndrome (γ-actin) (Appendix A). Arg256 is part of a pathogenic network, in which the so-called pathogenic helix (residues 113–126, Figure 1) and the C-terminal helix (residues 370–375) are interconnected. Arg256 modulates the lateral interaction of Lys113 with Glu195 in the adjacent monomer. Arg256 is also close to and likely to interact with Ile191 in H5. This network of interactions is thought to sense the binding of actin-binding proteins, and to communicate this to the rest of the actin molecule.

Pro70, found at the start of the sensor loop (His73), is mutated in every actin isoform except cardiac and γ-smooth actin. Mutation to Leu in β-actin causes Baraitser-Winter syndrome, and in γ-actin, causes Ocular coloboma [23,61]. A second mutation to Ala in β-actin also causes Baraitser-Winter syndrome [62]). Mutation to Arg in skeletal α-actin causes congenital myopathy [47] and to Gln in smooth α-actin causes thoracic aortic aneurisms [55]. The sensor loop that contains this residue is thought to function as a switch, linking changes in the nucleotide site to structural transitions in SD2, and in particular in the D-loop. Mutations would be expected to affect this sensor function and associated changes in structure in the D-loop, with downstream effects of filament stability.

Finally, mutations in Gly268 in the hydrophobic plug (FIGM) are found in four out of six isoforms. The hydrophobic plug is important for one of only two lateral interactions between actin monomers in the filament, in which it interacts with the D-loop. Gly268 interacts specifically with His40 as described above. This residue is mutated to Arg in both skeletal and smooth α-actin, causing nemaline myopathy [52,63] and aortic disease [64], respectively, and two further mutations (to Asp [65] or Cys [63]) in skeletal α-actin cause nemaline myopathy. The equivalent mutation in γ- and β-actin causes deafness [27] and Baraitser-Winter syndrome [66], respectively. Mutation of Gly268 is likely to weaken the lateral interaction with adjacent monomers, destabilizing the actin filament. Indeed, it is interesting that both His40 and Gly268 are mutated in multiple actin isoforms.

This analysis also shows some interesting differences in the positions of the mutations in each isoform. One of these is the lack of mutations in the region encompassing residues 220–230, just after the Thr-rich loop and H6, in isoforms other than skeletal and cardiac α-actin. This region has been identified as part of the actin binding interface for nebulin [67]. Nebulin is a large (~800 kDa) protein expressed in skeletal muscle that extends along the thin filament, binding to actin and tropomyosin and other sarcomeric proteins [68]. It contains a C-terminal SH3 domain, located in the Z-disc and 178 nebulin repeats. Recent data suggest that it stiffens the thin filament and contributes to thin filament activation in skeletal muscle [69]. Mutations in nebulin are a common cause of nemaline myopathy [12]. Therefore, in skeletal muscle, mutations in this region could disrupt the binding of nebulin to actin in the thin filament, weaken the thin filament and, thus, cause disease.

In cardiac muscle, nebulette and not nebulin is expressed. Nebulette is a member of the nebulin family of actin-binding proteins. Its N- and C-terminal regions are similar to those of nebulin, but it only has 32 nebulin repeats [68], is much smaller (~120 kDa) than nebulin and is not large enough to extend along the length of the thin filament. It is found in the Z-disc, and either projects a short distance along the filament from the Z-disc, interacts with desmin (as does nebulin) [70] and/or crosslinks actin filaments from adjacent sarcomeres within the Z-disc (reviewed in [71]). Mutations in nebulette have been linked to heart disease (HCM and DCM (reviewed in [71])). Assuming nebulette interacts with actin in a similar way to that of nebulin, mutations in actin in this region will also affect its ability to bind to cardiac α-actin, with potential downstream effects on actin filament stability.

A detailed knowledge of the structure of actin is important in predicting the precise effects of mutations. Indeed, the analysis presented here suggests that many mutations are likely to affect actin polymerization. However, some mutations will also affect the interaction of actin with its many other interacting proteins [1]. Other specific mutations can affect post-translational modifications [72] important for actin polymerization.

Three different post-translational modifications of actin have been linked to human cancers. First, Asp3 of β- (but not γ-) actin is arginylated by the enzyme ATE1 (arginyl-tRNA-protein transferase 1, which acts after the first two residues are removed [73]. A lack of arginylation reduces actin polymerization and its interaction with actin binding proteins [74]. Second is the specific acetylation of the N-terminal residues of β-actin and γ-actin by NAA80 (N-alpha acetyltransferase 80). A lack of NAA80 increases the numbers of actin filaments [75]. Third is the MICAL-mediated oxidation of Met44 and Met47 mentioned above. Levels of ATE1 are reduced in some human cancers and mutations in both NAA80 and ATE1 in cancer cell databases have also been reported. This suggests that both these enzymes could directly contribute to abnormal cell behaviour and metastasis through their effects on actin dynamics [2]. The depolymerization of F-actin initiated by MICAL is enhanced by the Abl non-receptor tyrosine kinase, and Abl is upregulated in several cancers (reviewed in [76]), suggesting a more indirect but important role of MICAL.

## 4. Mutations in Cardiac α-actin that Affect Myosin Interaction with Actin, Directly or Indirectly

Although the effects of mutations can be predicted, there remains a considerable challenge in experimentally determining their effects. This challenge is exemplified by considering a few examples of mutation in cardiac α-actin predicted to interact with myosin, and how they cause disease. These include mutations at the N-terminal region of actin, in the outer region of the D-loop and in subdomain 3 (between residues 311 and 335). The outer region of the D-loop is part of the ‘Milligan’ contact, in which myosin loop 3 (H551-G576) in the L50 domain interacts with actin SD1 and the D-loop of the adjacent actin subunit [77,78]. The precise mechanism by which myosin binds to actin can be somewhat variable between different myosin isoforms (e.g., compare NM2C and Myo6 [6,35]). However, each myosin primarily binds to residues in SD1, with some interactions in SD2 (such as the D-loop) of the adjacent actin, and SD3. Thus, mutations in residues within SD1 have the potential to disrupt myosin binding. For tropomyosin, key residues are D25 and the triad of residues, K326, K328 and R147, which interact with tropomyosin in the blocked (off) state of actin (reviewed in [79], and see Figure 1).

A relatively well understood mutation in cardiac α-actin is E99K (Glu99Lys), which causes HCM [80]. It is found in a region of actin thought to be involved in binding to the lower 50K domain of the myosin motor. E99K decreases the overall negative charge of the binding site within actin, and thus, weakens the acto-myosin interaction. Although one report suggested that E99K does not fold as well as wild type actin [81], a second report [82] showed that it does. Our lab has expressed this mutation as an eGFP-cardiac α-actin fusion construct in adult cardiomyocytes, using an adenoviral expression system [83] and found that it incorporates normally into the muscle sarcomere (Appendix A). In contrast, a recent study using C-elegans [84] reported that this mutation does form some aggregates, but when expressed in mouse hearts, this is not the case [85]. However, levels of expression are likely to affect these data, with higher expression levels more likely to lead to aggregates. Moreover, eGFP fused to actin can affect filament incorporation. For example, we found that eGFP-fused to the C-terminus of cardiac α-actin did not work well compared to eGFP-fused to the N-terminus (unpublished observation). Other work has also reported that the position of tags on actin can affect its properties [86,87]. Experimentally, myosin has been shown to bind more weakly to purified E99K F-actin and in vitro motility assays show that myosin moves both actin filaments [82] and reconstituted actin filaments (containing tropomyosin and troponin) [88] more slowly than wild type. Thus, the experimental data suggest that this mutation does not strongly affect the ability of actin to incorporate into thin filaments; however, once incorporated, this mutation affects the force output by reducing strong binding of myosin to actin, leading to compensatory hypertrophy.

A331P (Ala331Pro), which causes DCM [80], is in a region of actin that interacts with the cardiomyopathy loop in myosin [4,35] and with the tropomyosin binding site in the absence of Ca^2+^ [89]. However, rather than interfering with myosin binding directly, this mutation seems to do this indirectly by modulating the behaviour of tropomyosin. Recent work, using recombinant A331P cardiac α-actin, expressed using the baculovirus/insect cell expression system, showed that it polymerises faster than wild type actin, but in reconstituted thin filaments, there is a weaker interaction between myosin and actin [90]. This mutation was suggested to affect the interaction of tropomyosin with nearby residues, and, in particular, residues D25, R28 and P33, which form a bulge that defines the position of tropomyosin on actin in the ‘off’ (or blocked) state. A331P could increase the potential for tropomyosin to inhibit the acto-myosin interaction, by reducing the likelihood of movement of tropomyosin from its position in the ‘off’ state, to the ‘on’ state, thus explaining the decreased myosin interaction with the reconstituted thin filament. A331P also decreased the binding affinity for the C0C2 subunits of myosin binding protein C (MYBPC) to actin [91]. C0–C1 interact with both actin and tropomyosin to regulate contraction. In agreement with these findings, in unpublished work from our lab we found that eGFP-A331P was able to incorporate into muscle sarcomere (Appendix A) and while we did not find a significant effect on contraction (unpublished observation), this may be dependent on the levels of expression.

E361G (Glu361Gly), an HCM causing mutation [80], is close to the C-terminal myosin binding region of actin. However, it seems to exert its effect by indirectly affecting the binding of myosin to actin, through its effects on the Ca^2+^ sensitivity of the thin filament. E361G incorporates into thin filaments in the heart muscle in transgenic mice [92]. Our unpublished work also shows that eGFP-E361G incorporates normally into thin filaments in isolated cardiomyocytes, when expressed using an adenoviral system (Appendix A) and we also found it did not affect cardiomyocyte contraction (unpublished data). Thin filaments isolated from the transgenic mice, with E361G expressed at ~50% of the total actin show normal myosin driven motility in in vitro motility assays [92] and the mice have a very mild phenotype. However, cardiac contraction in these mice is not sensitive to phosphorylation of troponin I. The N-terminal peptide sequence of cardiac troponin-I is phosphorylated by protein kinase A (PKA) in response to β1-adrenergic signalling (reviewed in [93]). Through its interaction with troponin C, this decreases the Ca^2+^ binding affinity of troponin C, thus increasing the rate of Ca^2+^ dissociation and allowing the rate of twitch relaxation to increase. In turn, this allows the heart rate to increase, and increase force output [92]. Thus, the E361G mutation in actin uncouples the normal relationship between β1 adrenergic signalling, Ca^2+^ sensitivity and troponin I phosphorylation.

Finally, the mutation Arg312His (R312H), which causes DCM [17], is also likely to indirectly affect myosin binding to actin, by affecting the actin–tropomyosin interaction [88]. The myosin-driven velocity of reconstituted actin R312H filaments in in vitro motility assays, is reduced at high Ca^2+^ but increased at low Ca^2+^ concentrations compared to wild type actin [88]. However, strong binding of myosin to the mutant actin was unaffected. Thus, the observed changes in motility and Ca^2+^ sensitivity could be explained by an interaction of this region of actin with tropomyosin. In contrast, work by others has shown that R312H actin is less stable, less well able to polymerise, releases phosphate from its nucleotide site at a faster rate compared to wild type actin [94] and has a decreased actin-activated myosin ATPase linked to its reduced stability [95]. Both studies used recombinant actin produced from insect cells. In our unpublished work (Appendix A), we found that eGFP-R312H was able to incorporate into muscle sarcomeres in isolated adult rat cardiomyocytes and found no effect on contractility (unpublished observation). Given that the mutant actin appears to be able to incorporate into thin filaments in cells, a change to Ca^2+^ sensitivity, mediated through troponin/tropomyosin, may be more likely to account for the effects of this mutation.

## 5. Conclusions

The studies of a small number of mutations in cardiac α-actin discussed in the previous section demonstrate the challenges of trying to understand the effects of these mutations. Perhaps it is not surprising then that despite the large numbers of actin mutations reported, only a small number have been characterised in detail. A wide range of assays, both in vivo and in vitro, are generally needed to fully understand the effects of these mutations. Studies in vitro require purified actin and additional proteins (e.g., myosin, tropomyosin, troponin). This is not straightforward. Actin cannot be expressed and purified from *Escherichia. coli*, as it needs chaperones to fold correctly. However, it can be expressed and purified successfully using the Sf9/baculovirus system [82,86,96]. Studies in vivo require transgenic animals or human samples. The effects of mutations in intact skeletal fibres from humans or mouse models have also been characterised, as described above. Further examples include the demonstration that Phe352Ser increases contractile function [97], Asp286Gly prevents strong myosin binding [98] and His40Tyr inhibits co-polymerisation of wild type and mutant actin isoforms, and affects conformational changes in actin during contraction [99]. The use of model organisms, such as the indirect flight muscle of *Drosophila melanogaster*, has also proved useful [100]. This approach brings the advantage that the actin can be expressed on a null background, and can either be expressed as heterozygous (alongside a wild type copy of the gene) or homozygous, at levels typical of normal expression levels, and effects can be assessed in a mature muscle fibre. For example, this approach has shown that Asp292Val affects the regulation of contraction [100] through its interaction with tropomyosin [4].

Each of these systems has its own advantages and disadvantages. For example, in vitro expression systems do work but may only produce small amounts of actin [82,101]. The use of mouse models allows analysis of effects on adult skeletal fibres or the heart, but they are time consuming to make individual mutations and expensive to maintain mouse colonies. Human biopsies can be hard to obtain, and this research needs ethical approval. The use of *Drosophila Melanogaster* brings several advantages, however there are differences between the overall structure of its indirect flight muscle and the skeletal or cardiac muscles in humans.

In conclusion, disease-causing mutations are present in all six actin isoforms, and yet, the effects of many of these have not been well explored, particular those in smooth and non-muscle actin isoforms. While the effects of these mutations can be predicted from their positions in the structure, these predictions still need to be tested experimentally, to understand their potentially complex effects. The presence of mutations in similar regions of the sequence across the different actin isoforms, and their likely effects on filament stability, suggests it may well be worth exploring simple mammalian expression systems to analyse the effects of these mutations. CRISPR-based gene engineering approaches to directly edit the gene, while avoiding the use of tags, would allow analysis of actin dynamics in live cells, followed by purification of the expressed actin, to complement the many in vitro and in vivo assays already being used. The exploration of the role of PTMs in cancer biology is underexplored. It will be exciting to see the outcome of this type of research in the future.

## Figures and Tables

**Figure 1 ijms-21-03371-f001:**
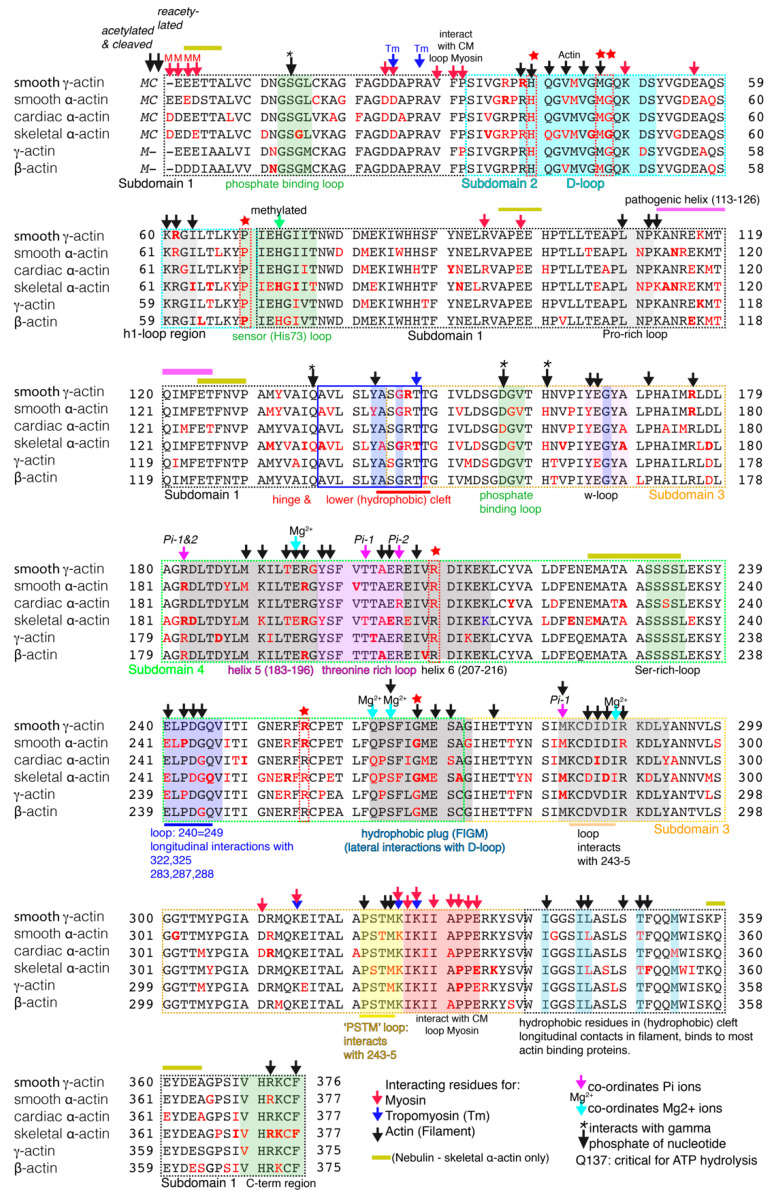
Annotated alignment of the six actin sequences for *Homo Sapiens* showing the positions of mutations. All six sequences were retrieved from UNIPROT (P63267: smooth γ-actin, P60709: β-actin: P68133: skeletal α-actin, P63261: γ-actin; P60382: cardiac α-actin, P62736: smooth α-actin). The first one–two residues are typically acetylated and cleaved, such that the third residue is the first residue in the expressed sequence. The positions of key interaction regions in the sequences is indicated together with positions of mutated residues (red font) in each sequence. Residues in which mutations are found in four out of the six isoforms are indicated by a red star. See text for more details. Numbering in the text refers to the residue number for α-actin isoforms. Mutations were retrieved from the Human Genome Mutation Database (HGMD).

**Figure 2 ijms-21-03371-f002:**
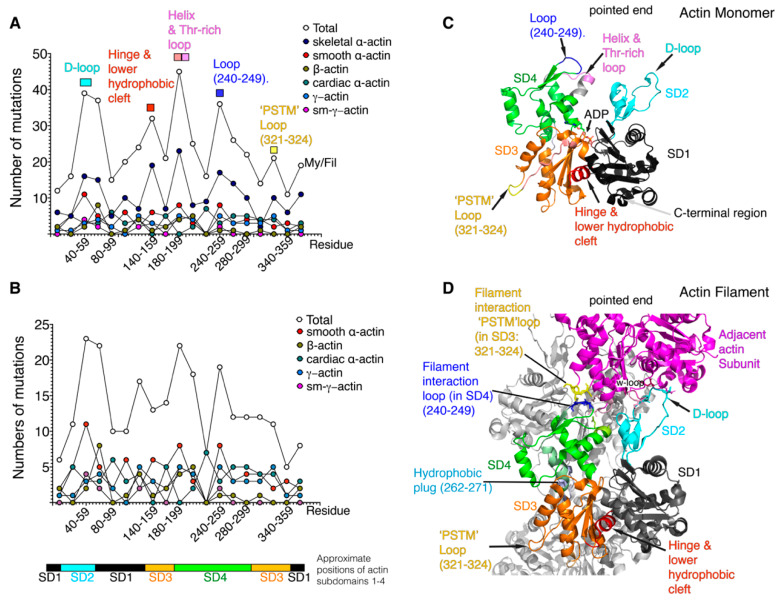
Analysis of actin mutations in all six *Homo sapiens* actin isoforms. Numbers of mutations in residues 1–19, 20–39 and so on, were summed and plotted as shown across the whole sequence for each individual isoform and for all isoforms (Total). (**A**) Panel A shows this analysis for each six actin isoforms, and the total for all six. The protein names are shown in the key. (**B**) Panel B is similar to A, except the mutations for ACTA1 were excluded. Mutations were taken from the HGMD (accessed November 2019, Appendix A). The approximate positions of the four subdomains found in the G-actin monomer (**C**) is additionally shown underneath panels A and B for reference. (**C**) The G-actin monomer is made up of four subdomains (denoted by black (subdomain-1, SD1), cyan (subdomain-2, SD2) orange (subdomain-3, SD3) and green (subdomain-4, SD4). Labels indicate regions of the structure that are hotspots for mutations. (**D**) The F-actin filament. One actin monomer is shown as in C, with the SD1-4 in black, cyan, orange and green, respectively. Positions where mutations are common are highlighted. The interaction between residues 243–245 in in SD4 (coloured blue), with residues 322–325 in SD3 (coloured yellow) in the adjacent actin subunit (coloured magenta) is indicated. Longitudinal contacts between the DNAse-1 loop (or D-loop; 38-52) and residues in the C-terminal region of the adjacent monomer (residues R372, F375), Pro-rich and w-loops are also shown. The helix coloured red (residues 135–146) in D is found in the lower hydrophobic cleft. The hydrophobic plug (residues 262–272, light blue) is important for lateral actin monomer contact between protofilaments. The regions of sequence for each subdomain have been defined as follows SD1 (residues 1–32, 70–144, and 338–372), SD2 (residues 33–69) SD3 (residues 145–180 and 270–337) SD4 (residues 181–269) [33].

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
