# Peer review of "Actin Mutations and Their Role in Disease"

_ijms, 2020, doi:10.3390/ijms21093371_

Round 1

Reviewer 1 Report

The manuscript by Baboolal and colleagues aims to provide an overview of the mutations that have been reported for the six actin genes, to relate individual mutations to specific functions,  and to describe approaches used for analyzing actin mutations. However, these objectives are not achieved.

The authors should apply a couple of basic rules to improve the impact of their work. They should reconsider the structure of their text and whether it makes sense to dedicate the final section to a superficial description of research approaches in the field. The use of clear and precise language and the systematic introduction of facts into the chain of arguments would be helpful. Finally, the authors should check the appropriateness and timeliness of some of the references.

Here are a few examples:

Genes are expressed but proteins are produced.

Have all 220 ACTA1 mutations been reported for the human gene? Do all 220 mutations result in single amino acid substitutions? Why do the authors no report on mutations that have a different outcome for any actin gene? Do all ACTA1 mutations resulting in single amino acid substitutions cause Nemaline Myopathy? Is there no other reference than [9]?

Cardiac actin is referred to as ACTAC1 and ACTC

The statement “ACTAC1 is one of eight disease 73 genes that contain missense mutations causing HCM, and one of 20 disease genes causing DCM [17, 74 18]” should precede the sentences  “HCM affects 1:500 to 1:250 people 68 and is a common cause of premature death in young adults” and “DCM is less common, affecting around 1:3000 people [16].”

Figure 2: The legibility of the labelling of panel D is marginal. Avoid yellow labelling in panels C and D.

Figure 3: The figure needs to be better integrated into the main text. The legend is insufficient.

Author Response

The manuscript by Baboolal and colleagues aims to provide an overview of the mutations that have been reported for the six actin genes, to relate individual mutations to specific functions,  and to describe approaches used for analyzing actin mutations. However, these objectives are not achieved.

The authors should apply a couple of basic rules to improve the impact of their work. They should reconsider the structure of their text and whether it makes sense to dedicate the final section to a superficial description of research approaches in the field. The use of clear and precise language and the systematic introduction of facts into the chain of arguments would be helpful. Finally, the authors should check the appropriateness and timeliness of some of the references.

Thank you for the very useful comments.  We have been through the paper, and tried to address these criticisms.

Here are a few examples:

Genes are expressed but proteins are produced.

Apologies for not making this clear, we have tried to be more consistent throughout.

Have all 220 ACTA1 mutations been reported for the human gene? Do all 220 mutations result in single amino acid substitutions? Why do the authors no report on mutations that have a different outcome for any actin gene? Do all ACTA1 mutations resulting in single amino acid substitutions cause Nemaline Myopathy? Is there no other reference than [9]?

Yes, of 222 mutations reported for ACTA1, 208 were missense.  The review is focussed on missense mutations only as these are typically the most common (e.g. over 90% are missense for ACTA1) – and the plot in figure 1 is only suitable for missense mutational analysis. We’ve now provided a table (supplementary Table 1) that lists all the missense mutations and their positions for each of the 6 actin isoforms, the diseases they are associated with and the accompanying reference. We’ve also added in one additional review to [9], and made clear that these are reviews.

Cardiac actin is referred to as ACTAC1 and ACTC

Apologies – we’ve corrected this mistake

The statement “ACTAC1 is one of eight disease 73 genes that contain missense mutations causing HCM, and one of 20 disease genes causing DCM [17, 74 18]” should precede the sentences  “HCM affects 1:500 to 1:250 people 68 and is a common cause of premature death in young adults” and “DCM is less common, affecting around 1:3000 people [16].”

We have re-ordered this paragraph

Figure 2: The legibility of the labelling of panel D is marginal. Avoid yellow labelling in panels C and D.

We have changed the yellow labelling to orange and improved the labelling

Figure 3: The figure needs to be better integrated into the main text. The legend is insufficient.

We have now incorporated much more extensive discussion of the mutations shown in this figure in the text and to reduce the prominence of Figure 3, we have moved it into Supplementary (Supplementary Fig. 1), and now provide an extended figure legend.

Reviewer 2 Report

In this review the group of Professor Peckham identifies the hotspots for mutations in six genes coding for actin isoforms. This is an important work bringing new knowledge about actins’ biology. However, this work has some flaws, which should be addressed by the Authors.

Major issues.

  1. This work focuses mainly on mutations found in genes coding for muscle-specific actin isoforms.
  2. There is nothing about mutations found in actins’ genes in tumors.
  3. There should be tables prepared presenting all up to date identified mutations for every actin gene with information about related disease and references related to given mutation (identification/functional studies etc.). This would be a genuine contribution to the field. Such tables could be presented in supplementary information.
  4. One of five chapters refers to mutations in positions crucial for actin:myosin interactions. I understand that the Authors are particularly interested in that, however not only myosins bind actin. Especially, that the Authors show in Fig. 1 residues important for tropomyosin binding. How about tropomodulins, gelsolin etc.?
  5. Figure 2 identifies the hotspots for mutations in actin genes, which are later on described in the text. The Authors draw general conclusions on the basis of plotted values for all actins taken together. This is quite informative, but taken into consideration that the huge majority of mutations was identified for muscle-specific actin isoforms, some interesting and potentially important data was neglected. For instance it is not discussed that there haven’t been identified yet any mutations for ACTC1 in one of the identified hotspots – Helix & Thr-rich loop. Whereas for the same gene and ACTA1 but not for the rest of actin genes there were detected 7 and 8 mutations in the region around 220-230 residues, respectively. And how about a difference in mutations frequency between ACTB and ACTG1 in subdomains 3 and 4? These observations should be as well discussed by the Authors as they can be meaningful.

Minor issues.

  1. The first paragraph of Introduction is almost a repetition of Abstract.
  2. The six actin genes are described in the Review only in relation to humans. That’s why the way of writing genes names should be changed from ACTB to ACTB. The names of Homo Sapiens genes should be written in italics. This is as well valid for figures. Moreover, it is not clear why the Authors in the text use appropriate nomenclature (ACTB, ACTG1, ACTG2, ACTA1, ACTA2, ACTC1) but in Fig. 1 they use ACTH, ACTA, ACTC, ACTS, ACTG and ACTB. It is confusing.
  3. Lines 61 and 63 – why there is ACTAC1? Shouldn’t it be ACTC1?
  4. Line 86 – Reference 24 is from 2004. Is there no recent literature/review about ACTG1 mutations?
  5. In Fig. 1’s description “red star” should be explained. There should be also a comment that the details are explained in the main text.
  6. Lines 140-144 – are there some disorders associated with mutations in position 48?
  7. Lines 164-210 – could the diseases associated with listed in these paragraphs mutations be named in the text? A reader has to look in the bibliography to know how mentioned mutations are manifested in patients. That would be a courtesy to the readers.
  8. In 2006 there was also another study published showing expression of actin in insect cells (DOI:10.1139/Y05-140). This should be cited as well.
  9. Lines 279-282 – a study from Manstein group (doi: 10.1007/s00018-012-1030-5.) should be also cited here as it is relevant for this Review. It concerns Tyr166Cys and Met305Leu mutations found in patients with cardiomyopathy.
  10. Fig. 3 – A short description of experimental conditions in the figure’s legend would be a courtesy to the reader. There should be included e.g. information about the time of incubation of cardiomyocytes with adenoviruses. Does the picture for E99K mutant show contracted sarcomeres? The staining pattern looks differently when compared to other mutants. Why there is shown a result for R312H mutant? Is it mentioned somewhere in the text? eGFP-band for E361G mutant looks stronger in comparison to other samples, so it can not be stated that the levels of studied actins are similar, especially that there is no loading control presented.
  11. Lines 253-258 – two works should be cited here as they show that actin “doesn’t like” tagging at the C-terminus (doi: 10.1251/bpo94 and DOI: 10.1091/mbc.10.1.135).
  12. A work from Dawson group concerning E99K mutation (doi: 10.1139/bcb-2014-0156.) should be discussed in the Review.
  13. There is missing a summary/conclusion paragraph.

Author Response

In this review the group of Professor Peckham identifies the hotspots for mutations in six genes coding for actin isoforms. This is an important work bringing new knowledge about actins’ biology. However, this work has some flaws, which should be addressed by the Authors.

We thank this reviewer for their valuable comments, and have tried to address each one as outlined below (in italics)

Major issues.

  1. This work focuses mainly on mutations found in genes coding for muscle-specific actin isoforms.  The review compares all genes, where mutations are found and the diseases they are found to cause. It would have to be a really extensive review to cover all the mutations in all the genes. As quite a bit of work has been done on skeletal and cardiac actin, we've actually chosen to focus on a small number of cardiac actin mutations simply to demonstrate that understanding their effects can be quite complex and involve many different approaches.  We've extended the discussion on this, to make this clearer.
  2. There is nothing about mutations found in actins’ genes in tumors. To our knowledge there is nothing published on missense mutations in actin that cause tumors. However, actin is extensively post-translationally modified, and mutations in genes that drive PTMs (NAA80 and ATE1) are linked to cancer. So, we have now included a discussion on that (lines 267-281).
  3. There should be tables prepared presenting all up to date identified mutations for every actin gene with information about related disease and references related to given mutation (identification/functional studies etc.). This would be a genuine contribution to the field. Such tables could be presented in supplementary information.  We have now completed such a table (supplementary table 1) for the missense mutations we analyse, with the associated references. Missense mutations are the predominant mutations for most of the actin isoforms.
  4. One of five chapters refers to mutations in positions crucial for actin:myosin interactions. I understand that the Authors are particularly interested in that, however not only myosins bind actin. Especially, that the Authors show in Fig. 1 residues important for tropomyosin binding. How about tropomodulins, gelsolin etc.?  We do make some reference to other proteins that bind actin (e.g. see second paragraph, in mutational analysis section). But we have chosen to focus on a small number of mutations, mostly in ACTC, that in this case, affect myosin binding as a way of demonstrating the different approaches used to try to understand the phenotype of these mutations. We have also expanded the text here quite a bit, as the effects of the specific mutations we choose to highlight also demonstrate that in fact several of these mutations have an indirect effect, and affect myosin binding through effects on tropomyosin/troponin. Again, the review would have to be much more extensive to cover all the interacting proteins in sufficient detail.
  5. Figure 2 identifies the hotspots for mutations in actin genes, which are later on described in the text. The Authors draw general conclusions on the basis of plotted values for all actins taken together. This is quite informative, but taken into consideration that the huge majority of mutations was identified for muscle-specific actin isoforms, some interesting and potentially important data was neglected. For instance it is not discussed that there haven’t been identified yet any mutations for ACTC1 in one of the identified hotspots – Helix & Thr-rich loop. Whereas for the same gene and ACTA1 but not for the rest of actin genes there were detected 7 and 8 mutations in the region around 220-230 residues, respectively. And how about a difference in mutations frequency between ACTB and ACTG1 in subdomains 3 and 4? These observations should be as well discussed by the Authors as they can be meaningful. 

    We have now included a discussion for some of the above. There is in fact one mutation in the Helix and Thr-rich region for ACTC, and for space regions we have not included a discussion on this. However, we have gone back and investigated the lack of mutations between 220-230 for genes other than ACTC and ACTA1, and now discuss this (lines 249-267). We also went back and looked at the difference in frequency of mutations between ACTB and ACTG1 in SD3 and 4, but we couldn’t see an obvious difference in frequency (about 50% of mutations in both ACTB and ATCG1 are in SD1 and 2, and 50% in SD3 & 4). So, we have not discussed this further.

Minor issues.

  1. The first paragraph of Introduction is almost a repetition of Abstract. We have rewritten the abstract
  2. The six actin genes are described in the Review only in relation to humans. That’s why the way of writing genes names should be changed from ACTB to ACTB. The names of Homo Sapiens genes should be written in italics. This is as well valid for figures. Moreover, it is not clear why the Authors in the text use appropriate nomenclature (ACTB, ACTG1, ACTG2, ACTA1, ACTA2, ACTC1) but in Fig. 1 they use ACTH, ACTA, ACTC, ACTS, ACTG and ACTB. It is confusing. ACTH,ACTA etc are the reference names for these proteins in UNIPROT. However, we agree these are confusing, so we have removed these from Figure 1 requested. Gene names are now in italics.
  3. Lines 61 and 63 – why there is ACTAC1? Shouldn’t it be ACTC1? – apologies – now corrected
  4. Line 86 – Reference 24 is from 2004. Is there no recent literature/review about ACTG1mutations? This reference is one of the first references to report mutations in ACTG1, which is why we cited this one at this point in the review. Ref 29 (in this paragraph, line 108)  is a more recent summary of ACTG1 mutations.
  5. In Fig. 1’s description “red star” should be explained. There should be also a comment that the details are explained in the main text. Apologies, this has now been done.
  6. Lines 140-144 – are there some disorders associated with mutations in position 48? Yes – now added (and new supplementary table 1 provides the details of the mutations, and related references)
  7. Lines 164-210 – could the diseases associated with listed in these paragraphs mutations be named in the text? A reader has to look in the bibliography to know how mentioned mutations are manifested in patients. That would be a courtesy to the readers.  The diseases are now named in the text, and  the new supplemental table also summarises the missense mutations and associated diseases.
  8. In 2006 there was also another study published showing expression of actin in insect cells (DOI:10.1139/Y05-140). This should be cited as well. This reference is now included
  9. Lines 279-282 – a study from Manstein group (doi: 10.1007/s00018-012-1030-5.) should be also cited here as it is relevant for this Review. It concerns Tyr166Cys and Met305Leu mutations found in patients with cardiomyopathy.  This reference is now included.
  10. Fig. 3 – A short description of experimental conditions in the figure’s legend would be a courtesy to the reader. There should be included e.g. information about the time of incubation of cardiomyocytes with adenoviruses. Does the picture for E99K mutant show contracted sarcomeres? The staining pattern looks differently when compared to other mutants. Why there is shown a result for R312H mutant? Is it mentioned somewhere in the text? eGFP-band for E361G mutant looks stronger in comparison to other samples, so it can not be stated that the levels of studied actins are similar, especially that there is no loading control presented.We’ve moved this figure into the supplemental data, as it does somewhat detract from the rest of the paper, but does serve to illustrate an additional approach that can be used to analyse actin mutations, so we feel should be included.  The legend to this figure now includes the incubation time, and some more detail about the technique. E99K shows stronger actin staining at the Z-disk (also seen in the phallodin image). This does not necessarily mean the cells have contracted, as the sarcomere length is ~2 microns as expected. I’ve also included the protein gel run at the same time, and with the same samples as the blot, to show similar loading.We apologise for leaving out R312H, this was an oversight. We now discuss this mutation properly in the paper.
  11. Lines 253-258 – two works should be cited here as they show that actin “doesn’t like” tagging at the C-terminus (doi: 10.1251/bpo94 and DOI: 10.1091/mbc.10.1.135). These references are now included
  12. A work from Dawson group concerning E99K mutation (doi: 10.1139/bcb-2014-0156.) should be discussed in the Review. This reference is now included.
  13. There is missing a summary/conclusion paragraph. This is now supplied.

Round 2

Reviewer 1 Report

The authors have taken up the main points of criticism. However, it seems that some of the changes have been implemented somewhat hastily. A further thorough reworking of the text seems advisable.

Author Response

We've been through the paper, and tried to correct and improve as much as we can. Hopefully, the revised paper is now better. 

Reviewer 2 Report

I thank very much the Authors for addressing all issues raised by me. I guess preparing the requested table costed a lot of work. Thank you for that.

I have one more ask to the Authors. Please, write in the abstract and instroduction that you discuss in more detail mainly mutations causing muscle-related disease. The title is very broad and suggests that you focus on all actins.

Author Response

We have now added a sentence in abstract and introduction regarding this